# The scientific chaos phase of the great pandemic: A longitudinal analysis and systematic review of the first surge of clinical research concerning COVID-19

Till Adami[1,2]*, Markus Ries[1]

1 Pediatric Neurology and Metabolic Medicine, Center for Pediatrics and Adolescent Medicine, Faculty of Medicine, Heidelberg University, Heidelberg, Germany, 2 Thoracic Oncology, Thoraxklinik Heidelberg, University Hospital Heidelberg, Heidelberg, Germany

* till.adami@med.uni-heidelberg.de

**Data Availability Statement:** All relevant data are within the paper and its Supporting Information files.

## Abstract

### Background

Early stages of catastrophes like COVID-19 are often led by chaos and panic. To characterize the initial chaos phase of clinical research in such situations, we analyzed the first surge of more than 1000 clinical trials about the new disease at baseline and after two years follow-up. Our 3 main objectives were: (1) Assessment of spatial and temporal evolution of clinical research of COVID-19 across the globe, (2) Assessment of transparency and quality —trial registration, (3) Assessment of research waste and redundancies.

### Methods

By entering the keyword "COVID-19" we screened the International Clinical Trials Registry Platform of the WHO and downloaded the search output when our goal of 1000 trials was reached on the 1st of April 2020. Additionally, we verified the integrity of the downloaded data from the meta registry by comparing the data with each individual registration record on their source register. Also, we conducted a follow-up after two years to track their progress.

### Results

(1) The spatial evolution followed the geographical spread of the disease as expected, however, the temporal development suggested that panic was the main driver for clinical research activities. (2) Trial registrations and registers showed a huge lack of transparency by allowing retrospective registrations and not keeping their registration records up to date. Quality of trial registration seems to have improved over the last decade, yet crucial information still was missing. (3) Research waste and redundancies were present as suggested by discontinuation of trials, preventable flaws in study design, and similar but uncoordinated research topics operationally fragmented in isolated silo-structures.

**Funding:** The authors received no specific funding for this work.

**Competing interests:** Markus Ries is a guest editor of the call for papers "Prediction and Mitigation of Natural Hazards". This does not alter our adherence to PLOS ONE policies on sharing data and materials.

## Conclusion

The scientific response mechanism across the globe was intact during the chaos phase. However, supervision, leadership, and accountability are urgently needed to prevent research waste, to ensure effective structure, quality, and validity to ultimately break the "panic-then-forget" cycle in future catastrophes.

## Introduction

In December 2019 a new respiratory disease of unknown cause was detected in Wuhan City, Hubei Province, China. A novel coronavirus, the severe acute respiratory syndrome coronavirus-2 (SARS-CoV-2), was identified as the cause of this unexpected outbreak. Due to its vast spread and global impact, the WHO declared this outbreak "a public health emergency of international concern" as early as of the 30[th] of January 2020, more than a month prior to being officially declared a pandemic on March 11, 2020 [1, 2].

The COVID-19 pandemic was a rapidly evolving situation, as was the scientific response: so far, the WHO registered 15834 trials on their clinical trials registry platform related to COVID-19 (accessed on 29[th] of June 2023) [3, 4]. The vast amount of emerging clinical research suggested that the scientific response mechanism across the globe was intact. Nevertheless, analogue to other catastrophic events such as mass casualty incidents (MCI), "chaos in catastrophic events impedes decision making and therefore hinders the scientific progress and thus effective treatment of patients" [5]. To surmount the initial chaos phase and to create order, appropriate response process structures are needed. In the scientific world of clinical research, defective process structure is often displayed by poorly designed studies, information overload and redundancies resulting in unnecessary costs and effort. Insufficient communication and non-transparency between researchers and institutions regarding planned, ongoing, and finished research is also a hindrance [6, 7].

Trial registration is a tool which is intended to help the public and scientific world by transparently displaying trial information and securing quality and validity. By minimizing publication, reproducibility and selective reporting bias, the value of research and publications is meant to be improved [8].

In analogy to the chaos phase in the first minutes of an MCI, we examined the "scientific chaos phase" of such a new pandemic situation by analyzing and reviewing the registration records of the first 1000 trials related to COVID-19 to detect possible flaws and opportunities for improvement in the future.

To assess lessons-learned, we developed the following 3 main objectives (Table 1):

## Materials and methods

This study was designed as a systematic review of the first 1000 clinical trial registration records registered in the WHO Registry Network related to COVID-19 to depict the scientific chaos phase of the pandemic. We followed the "Preferred Reporting Items for Systematic Reviews and Meta-Analyses" (PRISMA) guideline for reporting this study and included the checklist (S1 File). This review was not registered.

### Data sources and eligibility

The World Health Organizations International Clinical Trials Registry Platform (WHO ICTRP) is a registry network ("meta-registry") which gathers registration records from

**Table 1. Three main objectives with research questions, data sources, variables and hypotheses.**

| Main objectives | Questions | Data sources | Variables | Hypothesis |
|---|---|---|---|---|
| *(1) Assessment of Spatial and Temporal Evolution of Clinical Research of COVID-19 across the globe* | How was the geographical and temporal development of clinical research? | • WHO dataset<br>• ECDC<br>• United Nations population estimations | • Date of registration<br>• Origin country<br>• Source register<br>• Confirmed cases per country<br>• Population estimations per country | In the absence of a coordinating body, research follows the spread of the disease out of pragmatism, sense of urgency and panic |
| *(2) Assessment of Transparency and Quality—Trial Registration* | Is the data for each trial record transparent, accurate and up to date? How is the overall quality of trial registration? | • WHO data set<br>• Original registration record<br>• Follow-up assessment<br>• Publication search<br>• WHO TRDS and main criteria<br>• WHO primary registry guidelines | • Original recruitment status<br>• Latest recruitment status<br>• Date of first enrolment<br>• Estimated or actual study completion date<br>• Date of the last update | While it is extensively used in expert communities it is possible that there is little awareness, training, and rare usage of trial registries among a broad spectrum of health care professionals, therefore we hypothesized that there is a lack of tools for scientists to enable collaboration and communication |
| *(3) Assessment of research waste and redundancies* | Are there any signs of research waste and redundancies in the overall pattern of clinical trials? | • WHO dataset<br>• Original registration record<br>• Follow-up assessment | • Trial status (active or discontinued)<br>• Method of allocation<br>• Masking<br>• Study assignment<br>• If a control group was present<br>• Which type of control was established (standard operating procedures/standard of care vs. placebo vs. active comparator)<br>• Study phase<br>• Sample size | A surge like this is susceptible for poorly designed studies and similar study objectives resulting in a waste of resources and unnecessary exposure to clinical research because of a lack of central coordination and therefore supervision |

Tabular visualization of research questions, data sources, variables, and hypothesis for each of our three main objectives. WHO dataset, original search output from our WHO international clinical trials registry platform using the keyword "COVID-19"; ECDC, European Centre for Disease Prevention and Control [9]; UN, United Nations Department of Economic and Social Affairs Population Division World Population Prospects 2022 [10].

primary and partner registries and other data providers like ClinicalTrials.gov which meet the WHO registry criteria and the WHO Trial Registry Data Set (TRDS) [11–13].

To analyze the first surge of clinical research and cover the "scientific chaos phase" we decided to take a convenience sample of the first 1000 registered trials on the WHO ICTRP related to COVID-19. By entering the keyword "COVID-19" without any filters into the search portal of the WHO ICTRP (URL: https://apps.who.int/trialsearch/) we checked the number of registered trials on a regular basis. The goal of 1000 trials was reached on the 1st of April 2020. The search output was downloaded on the 22nd of April 2020 at 12:57:03 CET as a CVS-file and imported into Microsoft Excel with a total of 1528 eligible trials. The data used in this study is publicly accessible.

## Data extraction

The downloaded data set consists of 22 items for each trial (S2 File). The ICTRP also provides a hyperlink for each trial which leads to the original record of the source register so users can view additional information if necessary.

For our first objective, the spatial and temporal spread of clinical research across the globe, we additionally extracted the country of origin from each original record.

During the follow-up and for our second objective (the assessment of transparency and quality assessment of trial registration) and third objective (the assessment of redundancies and research waste) we extracted the following additional information from each original record: date of last update, current recruitment status, expected or actual recruitment completion date, detailed information about the study design (i.e., method of allocation, masking and assignment) and detailed information about the intervention (i.e., therapeutic agents being used, dosage, frequency and duration of application).

## First objective: Assessment of spatial and temporal evolution of clinical research of COVID-19 across the globe

To assess our first objective, we included all observational and interventional trials and used the sources and variables shown in Table 1. All trials were divided according to their country of origin, countries with less than 10 trials were summarized as "other". For better comparability and to determine if the differences in research activity are related to the population of a country or the incidence of COVID-19 cases we calculated the cumulative number of trials per 1 million inhabitants as well as the cumulative number of trials per 100 confirmed cases. The numbers of registered trials per 1 million inhabitants and per 100 confirmed cases were calculated by dividing the total number of registered trials from each continental region by the population respectively the confirmed cases of this region multiplying it with 1.000.000 respectively 100. Furthermore, to assess if research followed the spread of the disease due to missing coordination as we hypothesized, we graphically displayed the evolution of cumulative trials per region compared to the cumulative confirmed cases in this region. Additionally, to determine what impact official declarations from institutions like the WHO might have had on the research activity, we divided all trials in two groups depending on whether the trial was registered before or after COVID-19 was declared a pandemic. Lastly, we analyzed which registers were used most frequently and from which countries.

## Follow-up assessment

On the 11th of August 2022, more than 2 years after our initial data was collected, we initiated a follow-up search for each interventional trial to evaluate their progress and if the trials had been updated. From each original registration record, we manually extracted the latest recruitment status, overall trial status and the date of the last update. In this process we also identified if the investigated trials were active or discontinued. The discontinued trials were analyzed separately in our third objective (Assessment of Research Waste and Redundancies) for their reason of discontinuation and their investigated therapeutic agents.

Additionally, on the 18th of November 2022, more than 2,5 years after our initial data was collected, we performed a publication search for all active interventional trials, to find peer-reviewed and preprint articles. For this we used each individual Trial-ID and entered them into 4 major databases: LitCovid (https://www.ncbi.nlm.nih.gov/research/coronavirus/) & CochraneLibrary (https://www.cochranelibrary.com, with filter "all text") for peer-reviewed publications and MedRxiv (https://www.medrxiv.org) & preVIEW: COVID-19 (https://preview.zbmed.de, with filter "Multi") for preprint publications. Also, we verified if the publications or a summary of the results were linked to each trial protocol on their respective website as demanded by the WHO. Additionally, we calculated the time-to-publication in months by using the date of registration and the date on which the results were published first.

## Second objective: Assessment of transparency and quality–trial registration

To assess our second objective, we included all active interventional trials and used the sources and variables shown in Table 1. First and foremost, we compared the original recruitment status with the latest recruitment status. The original recruitment status was either labelled as "not recruiting" or "recruiting" for each record. The latest recruitment status was, depending on the source register, described as "not yet recruiting", "recruiting", "active, not recruiting", "enrolling by invitation", "completed" and sometimes not stated at all. Analogue to the original recruitment status "not yet recruiting" and "active, not recruiting" was coded as "not recruiting", "enrolling by invitation" was coded as "recruiting". With the date of first enrolment, the estimated or actual completion date and the latest recruitment status we checked if trials exceeded either their date of enrolment and were still labelled as "not recruiting" or exceeded their estimated date of completion and were still labelled as "recruiting" and therefore should have had been updated. In addition to the publication search we checked if the published trials were labelled as "completed". Also, we checked each record for retrospective registration by comparing the date of first enrolment with the date of registration. If the date of registration was in the past or on the date of first enrolment the trial was coded as "prospectively registered", otherwise it was coded as "retrospectively registered". Lastly, by comparing the date of the last update with the date of registration we checked if there were trials which never had been updated.

To assess the quality of the trial registrations, we screened each original registration record for completeness of the Trial Registration Data Set (TRDS, version 1.3.1) which currently contains 24 items [12]. The complete list of the 24 items including the explanation at which point an item was considered as missing or displayed insufficiently is shown in (S2 File).

## Third objective: Assessment of research waste and redundancies

To assess our third objective, we included all interventional trials and the used the sources and variables shown in Table 1. We first analyzed all discontinued trials for their reason of discontinuation and the rate of discontinuation per register. After that we analyzed the study design of all active interventional trials. Additionally, we identified all active and discontinued interventional trials which investigated therapeutic agents except those which investigated complementary or alternative medicine such as traditional Chinese medicine and trials which used any advanced medicinal products such as umbilical cord stem cells. After that we screened those trials for each therapeutic agent being investigated to determine how often each drug was tested, and which was tested most often. We divided the results into active and discontinued trials to determine how many trials originally planned on investigating those interventions.

## Statistical analysis

We used standard methods for descriptive statistics. All variables are either expressed as mean values with standard deviation, medians, and ranges or with absolute counts and percentages. Figs 4 and 5 are displayed as box-whiskers-plots. Temporal analyzations are displayed in months. Data analysis and figures were performed using SPSS (version 29, IBM) and Microsoft Excel.

## Results

### Trial selection

The trial selection is displayed in a flow diagram according to the PRISMA-guidelines (Fig 1). On the 1st of April our targeted convenience sample size of 1000 studies was exceeded. To

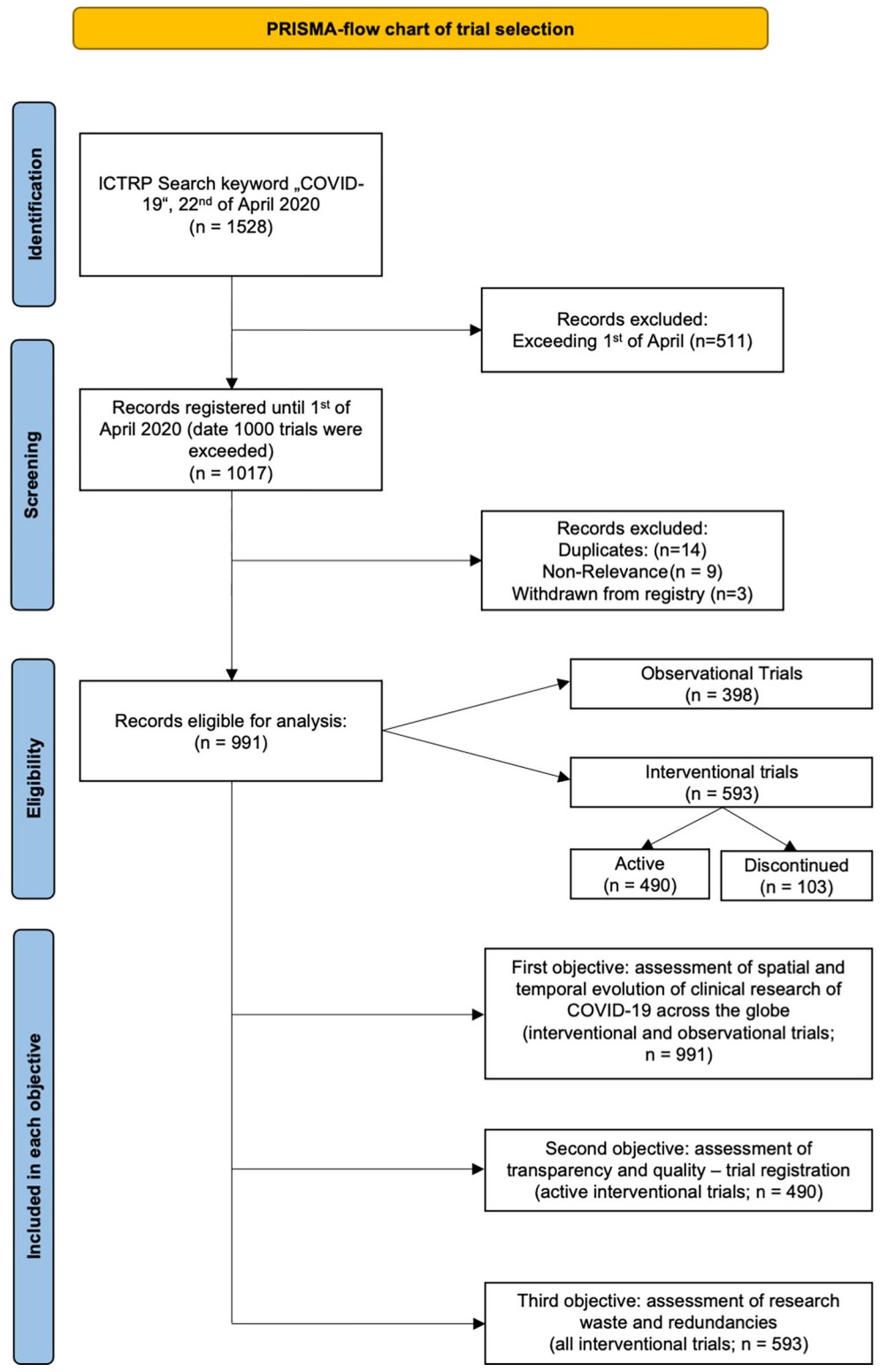

**Fig 1. PRISMA-flow chart of trial selection: Eligibility and inclusion process for each objective.**

avoid any sort of bias by falsely excluding any trials—since we could not identify which trial was the 1000[th]—we included all trials registered until that day, resulting in a total of 1017 eligible trials. After excluding trials not related to COVID-19, duplicates and trials withdrawn from the registry we ended up with 991 included trials. For our first objective we included all 991 trials. As the ICMJE and WHO consider trial registration to be mandatory only for interventional trials, we included only interventional trials for our follow-up search, our second objective (Assessment of Transparency and Quality Assessment–Trial Registration) and our third objective (Assessment of Redundancies and Research Waste) (n = 593). As we examined the trial registration process itself with these objectives, inclusion of observational trials would have led to false alteration of results as every information given about their trial registration can be considered voluntary [8]. During the follow-up we identified 103 discontinued interventional trials which will be analyzed separately in our third objective for their reason of discontinuation and their investigated therapeutic agents. Therefore, for our second objective (assessment of transparency and quality) we included all 490 active interventional trials and for our third objective (assessment of research waste and redundancies) we included all 593 interventional trials.

## First objective: Assessment of spatial and temporal evolution of clinical research of COVID-19 across the globe

The very first trial related to COVID-19 was registered on the 23[rd] of January. While there were only 15 trials registered in the last week of January, the numbers rapidly increased to 360 registered trials in February, rising to 568 trial registrations in March and 48 registrations only on the 1[st] of April, resulting in a total of 991 registered trials in less than 3 months. The first trials had their origins in China while the first trial outside of China was registered on the 5[th] of February 2020 in France. For a clearer demonstration of the differences in trial emergence, we graphically displayed the temporal evolution of trial registration in China compared to the rest of the World (Fig 2).

All in all, we identified a total of 39 different countries with 11 countries registering 10 or more trials. All countries with less than 10 trials will be summarized as "other". China had by far the most registered trials with a total of 663 (66,9%) of all 991 trials. Second most often was USA with 61 (6,2%) followed by France with 44 (4,4%), Iran with 40 (4,0%) and Italy with 24 (2,4%) trials. The remaining 159 trials originated in decreasing numbers from Germany, United Kingdom, Netherlands, Spain, Canada, Japan, and others. If calculated by trials per 1 million inhabitants, Netherlands (0,80), France (0,682) and China (0,465) were leading. Considering the number of trials per 100 confirmed cases China led with 0,806 trials per 100 cases, followed by Japan with 0,512 and Canada with 0,141 (Table 2).

If divided into before and after COVID-19 was declared a pandemic on the 11[th] of March 2020, we see that the distribution is nearly even with 485 (48,9%) registered before and 506 (51,1%) trials registered after the declaration. A majority of 461 (95,1%) of all 485 trials registered before the pandemic declaration originated from China while after the declaration only 201 (39,7%) of all 506 trials registered after the declaration originated from China. This is followed by the USA with 9 (1,9%) trials before and 52 (10,2%) trials after the declaration, France with 5 (1,0%), respectively 39 (7,7%) trials and Iran with all 40 trials (7,9%) registered after the pandemic declaration. Moreover, most of the trials originating from China were registered before the declaration with 69,6%, while most of the trials from other regions were registered after the declaration with the U.S. registering 85,2%, France registering 88,6% and Iran registering 100% of their trials after the declaration. In addition, we graphically depicted the temporal evolution of confirmed cases compared to registered trials. In this view, it appears that the

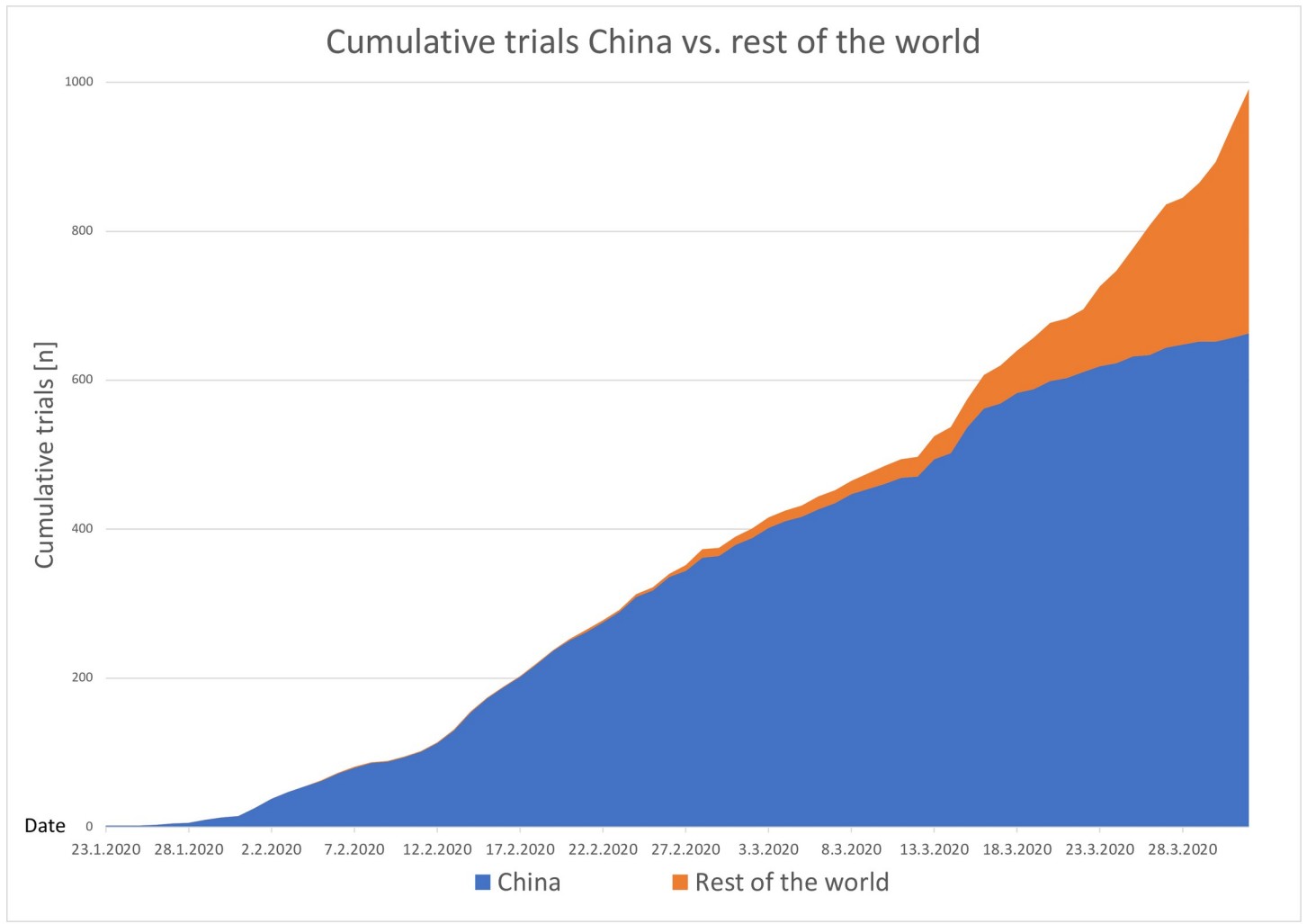

**Fig 2. Evolution of trial registrations in China vs. rest of the world.** X-axis: Timeline from 23rd January 2020 to 01st April 2020, Y-axis: Cumulative trials registered in the ICTRP with the keyword COVID-19 from China (blue) and the Rest of the World (orange). Data was extracted from the original dataset downloaded from the ICTRP and from the original registration record of each trial.

Chinese curve is already in stagnation for both cases and trials, whereas the curve of the other countries just start to rise (Fig 3, Table 2).

Overall, 12 different registries were used for registration: Chinese Trials Registry (CTR), ClinicalTrials.gov, Iranian Registry of Clinical Trials (IRCT), European Clinical Trials Registry (EU-CTR), Japan Primary Registry Network (JPRN), German Clinical Trials Registry (DRKS), Netherlands Trial Registry (NTR), International Standard Registered Clinical Trial Number registry (ISRCTN), Thailand Clinical Trials Registry (TCTR), Clinical Trials Registry India (CTRI), Australian and New Zealand Clinical Trials Registry (ANZCTR) and the Brazilian Registry of Clinical Trials (ReBEC). ChiCTR was used most frequently with 560 (56,5%) trials of which all originated from China. Clinicaltrials.gov was used second most often with a total of 325 (32,8%) trials of which 103 originated from China, 61 from the US, 31 from France, 24 from Italy and the remaining 109 trials in decreasing numbers from Canada, UK, Spain, Germany, Iran, Netherlands, and others. IRCT is third with 37 (3,7%) trials, which all originated from Iran, followed by EU-CTR with 26 (2,6%) registrations and JPRN with 10 (1,0%)

Table 2. Spatial and temporal analysis of all 991 observational and interventional trials.

| | | Country of origin (%) | | | | | | | | | | | |
| | CHN | USA | FRA | IRN | ITA | DEU | GBR | NLD | ESP | CAN | JPN | Other | World |
|---|---|---|---|---|---|---|---|---|---|---|---|---|---|
| Population estimates* | 1425893000 | 336998000 | 64531000 | 87923000 | 59240000 | 83409000 | 67281000 | 17502000 | 47487000 | 38155000 | 124613000 | 5556263000 | 7909295000 |
| **Registered trials [n]** | | | | | | | | | | | | | |
| Total | 663 (66,9) | 61 (6,2) | 44 (4,4) | 40 (4,0) | 24 (2,4) | 19 (1,9) | 15 (1,5) | 14 (1,4) | 12 (1,2) | 12 (1,2) | 10 (1,0) | 77 (7,8) | 991 (100) |
| January | 15 (100) | 0 (0%) | 0 (0) | 0 (0) | 0 (0) | 0 (0) | 0 (0) | 0 (0) | 0 (0) | 0 (0) | 0 (0) | 0 (0) | 15 (1,5) |
| February | 349 (96,9) | 6 (1,7) | 2 (0,6) | 0 (0) | 0 (0) | 0 (0) | 0 (0) | 0 (0) | 0 (0) | 0 (0) | 2 (0,6) | 1 (0,3) | 360 (36,3) |
| March | 293 (51,6) | 51 (9,0) | 36 (6,3) | 34 (6,0) | 24 (4,2) | 15 (2,6) | 12 (2,1) | 13 (2,3) | 10 (1,8) | 10 (1,8) | 7 (1,2) | 63 (11,3) | 568 (57,3) |
| 1st of April | 6 (12,5) | 4 (8,3) | 6 (12,5) | 6 (12,5) | 0 (0) | 4 (8,3) | 3 (6,3) | 1 (2,1) | 2 (4,2) | 2 (4,2) | 1 (2,1) | 13 (27,1) | 48 (4,8) |
| Before declaration | 461 (69,5) | 9 (14,8) | 5 (11,4) | 0 (0) | 0 (0) | 0 (0) | 1 (6,7) | 0 (0) | 1 (8,3) | 1 (8,3) | 4 (40,0) | 3 (3,9) | 485 (48,9) |
| After declaration | 202 (30,5) | 52 (85,2) | 39 (88,6) | 40 (100) | 24 (100) | 19 (100) | 14 (93,3) | 14 (100) | 11 (91,7) | 11 (91,7) | 6 (60,0) | 74 (96,1) | 506 (51,1) |
| Per 1 million inhabitants | 0,465 | 0,181 | 0,682 | 0,455 | 0,405 | 0,228 | 0,223 | 0,80 | 0,253 | 0,315 | 0,08 | 0,014 | 0,125 |
| Per 100 confirmed cases | 0,806 | 0,032 | 0,084 | 0,09 | 0,023 | 0,028 | 0,044 | 0,111 | 0,011 | 0,141 | 0,512 | 0,044 | 0,112 |
| **Trials per register [n]** | | | | | | | | | | | | | |
| ChiCTR | 560 (100) | 0 (0) | 0 (0) | 0 (0) | 0 (0) | 0 (0) | 0 (0) | 0 (0) | 0 (0) | 0 (0) | 0 (0) | 0 (0) | 560 (56,5) |
| ClinicalTrials.gov | 103 (31,7) | 61 (18,8) | 31 (9,5) | 3 (0,9) | 24 (7,4) | 8 (2,5) | 11 (3,4) | 3 (0,9) | 11 (3,4) | 12 (3,7) | 0 (0) | 58 (17,8) | 325 (32,8) |
| IRCT | 0 (0) | 0 (0) | 0 (0) | 37 (100) | 0 (0) | 0 (0) | 0 (0) | 0 (0) | 0 (0) | 0 (0) | 0 (0) | 0 (0) | 37 (3,7) |
| EU-CTR | 0 (0) | 0 (0) | 11 (42,3) | 0 (0) | 0 (0) | 2 (7,7) | 1 (3,8) | 2 (7,7) | 1 (3,8) | 0 (0) | 0 (0) | 9 (34,6) | 26 (2,6) |
| JPRN | 0 (0) | 0 (0) | 0 (0) | 0 (0) | 0 (0) | 0 (0) | 0 (0) | 0 (0) | 0 (0) | 0 (0) | 10 (100) | 0 (0) | 10 (1,0) |
| Other | 0 (0) | 0 (0) | 2 (6,1) | 0 (0) | 0 (0) | 9 (27,3) | 3 (9,1) | 9 (27,3) | 0 (0) | 0 (0) | 0 (0) | 10 (30,3) | 33 (3,3) |
| Total | 663 (66,9) | 61 (6,2) | 44 (4,4) | 40 (4,0) | 24 (2,4) | 19 (1,9) | 15 (1,5) | 14 (1,4) | 12 (1,2) | 12 (1,2) | 10 (1,0) | 77 (7,8) | 991 (100) |

CHN, China; USA, United States of America; FRA, France; IRN, Iran; ITA, Italy; DEU, Germany; GBR, Great Britain; NLD, Netherlands; ESP, Spain; CAN, Canada; JPN, Japan. ChiCTR, Chinese Clinical Trials Registry; IRCT, Iranian Registry for Clinical Trials; EU-CTR, European Clinical Trials Registry; JPRN, Japan Primary Registry Network. All numbers are total counts. Percentages of the rows "Before declaration" and "After declaration" refer to the respective value of row "Total". Percentages of the column "World" refer to the total 991 trials of the first row. The other percentages of each row refer to the respective value of the column "World". Data was extracted from the original dataset downloaded from the ICTRP and from the original registration record of each trial.

*UN, Word Population Prospects 2022 estimates as of 1st July 2021 [10].

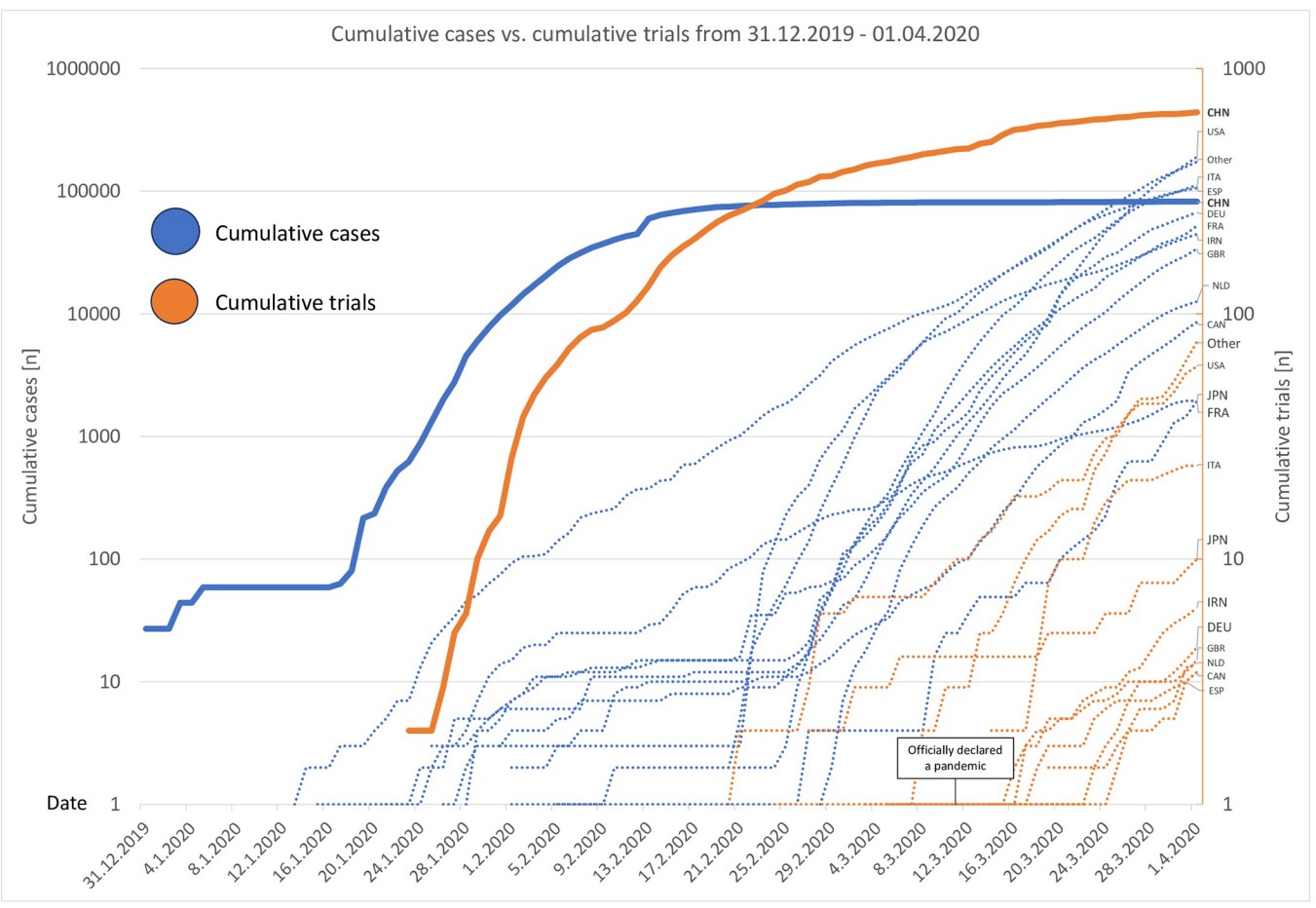

**Fig 3. Cumulative trials vs. cumulative cases from 31.12.2019–01.04.2020.** CHN, China; USA, United States of America; FRA, France; IRN, Iran; ITA, Italy; DEU, Germany; GBR, Great Britain; NLD, Netherlands; ESP, Spain; CAN, Canada; JPN, Japan. X-Axis: Timeline from the 31st of December 2019 to 01st of April 2020. Y-Axis: Logarithmic representation of the cumulative number of confirmed cases of COVID-19 (blue) compared to the cumulative number of registered trials regarding COVID-19 (orange) for each country. The numbers of trials were extracted from the ICTRP with the keyword COVID-19, the number of confirmed cases were extracted from publicly available data from European Center for Disease Control.

registered trials. The remaining 33 trials are spread over the remaining registers with all having less than 10 registrations each and therefore are summarized as "other" (Table 2).

## Follow-up assessment after 2 years

For the assessment of transparency in trial registration of our second objective we conducted a follow up search for all 593 interventional trials after more than two years during which we extracted the latest recruitment status, overall trial status and the date of the last update. Nearly half of those trials were registered with the ChiCTR (49,4%), more than a third were registered with ClinicalTrials.gov (37,1%), and the remaining were registered in decreasing order with the IRCT, the EU-CTR and the JPRN. Registers with less than 10 registrations each were summarized as "other". During the follow-up process we identified 103 (17,4%) of those trials were discontinued. The discontinued trials were separately analyzed in our third objective for their reason of discontinuation and their investigated therapeutic agents (Table 3).

**Publication pattern.** To assess the publication pattern of the 490 active interventional trials we conducted a publication search two and a half years after our initial data was collected. We identified a total of 139 trials (28,4%) which already published results after 2 years. 84,2%

**Table 3. Intervention trials per register and discontinuation of trials.**

| | Source Register (%) | | | | | |
|---|---|---|---|---|---|---|
| | **ChiCTR** | **ClinicalTrials.gov** | **IRCT** | **EU-CTR** | **Other[1]** | **Total** |
| **Interventional trials** | | | | | | |
| **in numbers n** | | | | | | |
| Total registered interventional trials | 293 | 220 | 37 | 26 | 17 | 593 |
| Total discontinued interventional trials | 39 | 48 | 0 | 13 | 3 | 103 |
| Total active interventional trials | 254 | 172 | 37 | 13 | 14 | 490 |
| Discontinuation rate per register in % | 13,3 | 21,8 | 0,0 | 50,0 | 17,6 | 17,4 |
| **Reason for discontinuation in numbers n** | | | | | | |
| Not stated | 33 (84,6) | 0 (0) | - | 6 (46,2) | 2 (66,7) | 41 (39,8) |
| Difficulties with enrolment | 5 (12,8) | 22 (35,8) | - | 3 (23,1) | 0 (0) | 30 (29,1) |
| New evidence/treatment guidelines | 0 (0) | 10 (20,8) | - | 2 (15,4) | 1 (33,3) | 13 (12,6) |
| Authority decision or recommendation | 0 (0) | 5 (10,4) | - | 0 (0) | 0 (0) | 5 (4,9) |
| Changed to expanded access study | 0 (0) | 5 (10,4) | - | 0 (0) | 0 (0) | 5 (4,9) |
| Logistics, staff, or resource issues | 0 (0) | 2 (4,2) | - | 1 (7,7) | 0 (0) | 3 (2,9) |
| Change of study design/protocol | 1 (2,6) | 2 (4,2) | - | 1 (7,7) | 0 (0) | 4 (3,9) |
| Planned termination criteria | 0 (0) | 1 (2,1) | - | 0 (0) | 0 (0) | 1 (1,0) |
| Futility | 0 (0) | 1 (2,1) | - | 0 (0) | 0 (0) | 1 (1,0) |

ChiCTR, Chinese Clinical Trials Registry; IRCT, Iranian Registry for Clinical Trials; EU-CTR, European Clinical Trials Registry. All numbers are total counts. Percentages for the section "Reason for discontinuation" refer to the respective value of the row "Total discontinued interventional trials". Data was extracted from the original dataset downloaded from the ICTRP and from the original registration record of each trial.

[1]Other registers include: JPRN, Japanese Primary Registry Network (n = 6); NTR, Netherlands Trial Registry (n = 3); ISRCTN, International Standard Registered Clinical Trial Number registry (n = 3); DRKS, German Clinical Trials Registry (n = 1); TCTR, Thailand Clinical Trials Registry (n = 1); ReBEC, Brazilian Registry of Clinical Trials (n = 1); ANZCTR, Australian and New Zealand Clinical Trials Registry (n = 1); CTRI, Clinical Trials Registry India (n = 1).

of those publications were already peer-reviewed and 15,8% were only available as preprint articles. Of those 139 trials only 18,7% registration records provided a link or a summary of those results on their original record on their respective source register. 3 (2,2%) trials have submitted a summary of their results to their registry, but those have not yet been published. Detailed results are shown in Table 4.

Additionally, we graphically displayed the time to publication calculated from the date of registration to the date the publications were first published. Peer-reviewed articles were published after a mean time of 11,69 months (SD: 7,78), a median time of 10,59 months, the 25 percentile was 4,66 months and the 75 percentile was 17,59 months. Preprint articles were published after a mean time of 4,12 months (SD: 2,53), a median time of 3,89 months, the 25 percentile was 2,1 months and the 75 percentile was 6,2 months (Fig 4).

## Second objective: Assessment of transparency and quality–trial registration

As stated above, the WHO and ICMJE consider trial registration to be mandatory only for interventional trials, therefore we excluded all observational trials (n = 398) for this objective, the assessment of the trial registration itself.

**Assessment of transparency.** To assess the transparency of trial registration we excluded all trials which were found as discontinued in the follow-up search (n = 103). The discontinued trials were analyzed separately for their reason of discontinuation and their therapeutic agents in the third objective. This resulted in 490 included trials and only 4 registers with more than

**Table 4. Publication pattern per register.**

| | ChiCTR (%) | ClinicalTrials.gov (%) | IRCT (%) | EU-CTR (%) | Other[1] (%) | Total (%) |
|---|---|---|---|---|---|---|
| **Total active interventional trials** | 254 (51,8) | 172 (35,1) | 37 (7,6) | 13 (2,7) | 14 (2,9) | 490 (100) |
| **Published Trials** | | | | | | |
| Total | 47 (18,5) | 74 (43,0) | 6 (16,2) | 6 (46,2) | 6 (42,9) | 139 (100) |
| Peer-Reviewed | 31 (66,0) | 69 (93,2) | 5 (83,3) | 6 (100) | 6 (100) | 117 (84,2) |
| Preprint | 16 (34,0) | 5 (6,8) | 1 (16,7) | 0 | 0 | 22 (15,8) |

ChiCTR, Chinese Clinical Trials Registry; IRCT, Iranian Registry for Clinical Trials; EU-CTR, European Clinical Trials Registry. All numbers are total counts. Percentages of the row "Total active interventional trials" refer to the total number of active trials (n = 490). Percentages of the row "Total" refer to the respective value of the row "Total active interventional trials". Percentages of the rows "Peer-reviewed" and "Preprint" refer to the respective value of row "Total". Data was extracted from the original dataset downloaded from the ICTRP and from the original registration record of each trial.

[1]Other registers include: JPRN, Japanese Primary Registry Network (n = 5); NTR, Netherlands Trial Registry (n = 2); ISRCTN, International Standard Registered Clinical Trial Number registry (n = 3); DRKS, German Clinical Trials Registry (n = 1); ReBEC, Brazilian Registry of Clinical Trials (n = 1); ANZCTR, Australian and New Zealand Clinical Trials Registry (n = 1); CTRI, Clinical Trials Registry India (n = 1).

10 registrations: ChiCTR (n = 254), ClinicalTrials.gov (n = 172), IRCT (n = 37) and EU-CTR (n = 13). The rest of 14 trials will be summarized as others. The results for this assessment are displayed in Table 5.

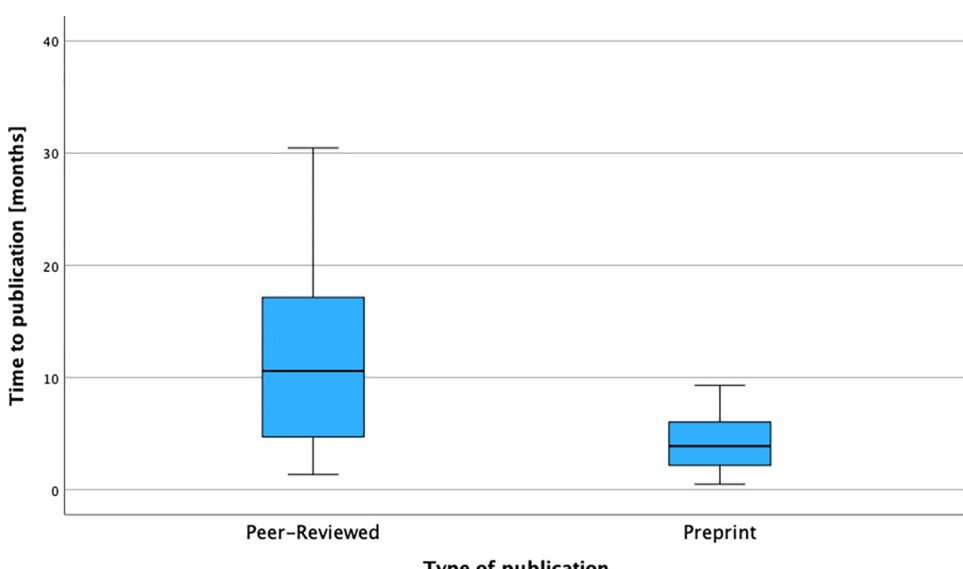

**Fig 4. Box-whiskers-plot of the time-to-publication divided into peer-reviewed and preprint articles.** X-axis: Type of publication divided into peer-reviewed and preprint articles, Y-axis: Time to publication expressed in months calculated as the period from the date of registration to the date of first publication. The publication search was conducted by entering each individual trial-ID into 4 major databases: LitCOVID, CochraneLibrary, MedRxiv and preVIEW:COVID-19. The date of registration was extracted from the original database.

**Table 5. Assessment of transparency–Analysis of the progress of active interventional trials.**

| | Source Register (%) | | | | | |
|---|---|---|---|---|---|---|
| | **ChiCTR** | **ClinicalTrials. gov** | **IRCT** | **EU-CTR** | **Other[1]** | **Total** |
| **Total active interventional trials** | 254 (51,8) | 172 (35,1) | 37 (7,6) | 13 (2,7) | 14 (2,9) | 490 (100) |
| **Prospective/Retrospective registration** | | | | | | |
| Prospective | 135 (53,1) | 132 (76,7) | 13 (35,1) | 12 (92,3) | 13 (92,9) | 305 (62,2) |
| Retrospective | **119 (46,9)** | **40 (23,3)** | **24 (64,)** | **1 (7,7)** | **1 (7,1)** | **185 (37,8)** |
| **Original recruitment status (time of registration)** | | | | | | |
| Recruiting | 145 (57,1) | 91 (52,9) | 23 (62,2) | 1 (7,7) | 10 (71,4) | 270 (55,1) |
| Not Recruiting | 109 (42,1) | 81 (47,1) | 14 (37,8) | 12 (92,3) | 4 (28,6) | 220 (44,9) |
| **Latest recruitment status (11 August 2022)** | | | | | | |
| Recruiting | 144 (56,7) | 46 (26,7) | 0 (0) | 0 (0) | 2 (14,3) | 192 (39,2) |
| Not Recruiting | 103 (40,6) | 44 (25,6) | 1 (2,7) | 0 (0) | 2 (14,3) | 150 (30,6) |
| Completed | 7 (2,7) | 82 (47,7) | 36 (97,3) | 6 (46,2) | 10 (71,4) | 141 (28,8) |
| Not Stated | 0 (0) | 0 (0) | 0 (0) | 7 (53,8) | 0 (0) | 7 (1,4) |
| **Trials exceeding enrolment date per recruitment status** | | | | | | |
| Recruiting | 144 (56,7) | 46 (26,7) | 0 (0) | 0 (0) | 2 14,3) | 192 (39,2) |
| Not Recruiting | **103 (40,6)** | **44 (25,6)** | **1 (2,7)** | 0 (0) | **2 (14,3)** | **150 (30,6)** |
| Completed | 7 (2,7) | 82 (47,7) | 36 (97,3) | 6 (46,2) | 10 (71,4) | 141 (28,8) |
| Not Stated | 0 (0) | 0 (0) | 0 (0) | 7 (53,8) | 0 (0) | 7 (1,4) |
| Total | 254 | 172 | 37 | 13 | 14 | 490 |
| **Trials exceeding estimated/actual completion date per recruitment status** | | | | | | |
| Recruiting | **139 (54,7)** | **38 (22,1)** | 0 (0) | 0 (0) | 0 (0) | **177 (36,1)** |
| Not Recruiting | 100 (39,4) | 32 (18,6) | 1 (2,7) | 0 (0) | 0 (0) | 133 (27,1) |
| Completed | 7 (2,8) | 81 (47,1) | 36 (97,3) | 6 (46,2) | 8 (57,1) | 138 (28,2) |
| Total | 246 (96,9) | 151 (87,8) | 37 (100) | 6 (46,2) | 8 (57,1) | 448 (91,4) |
| **Update history of trials** | | | | | | |
| Never Updated | **102 (40,2)** | 0 (0) | **16 (43,2)** | * | **1 (12,5)** | **119 (24,3)** |
| Within first week | 69 (27,2) | 13 (7,6) | 5 (13,5) | * | 0 (0) | 87 (17,8) |
| Within first month | 56 (22,0) | 28 (16,3) | 5 (13,5) | * | 0 (0) | 89 (18,2) |
| Within first half year | 22 (8,7) | 40 (23,3) | 8 (21,6) | * | 2 (14,3) | 72 (14,7) |

*(Continued)*

**Table 5.** (Continued)

| | Source Register (%) | | | | | |
|---|---|---|---|---|---|---|
| | ChiCTR | ClinicalTrials.gov | IRCT | EU-CTR | Other[1] | Total |
| More than a half year after registration | 5 (1,9) | 91 (52,9) | 3 (8,1) | * | 11 (78,6) | 110 (22,4) |
| **Never updated trials per recruitment status** | | | | | | |
| Recruiting | 54 (53,0) | 0 (0) | 0 (0) | * | 0 (0%) | 54 (45,4) |
| Not recruiting | 47 (46,1) | 0 (0) | 1 (6,3) | * | 1 (100) | 49 (41,2) |
| Completed | **1 (0,9)** | 0 (0) | **15 (93,8)** | * | 0 (0) | 16 (13,4) |
| **Published trials per recruitment status** | | | | | | |
| Total | **47 (18,5)** | 74 (43,0) | **6 (16,2)** | 6 (46,2) | 6 (42,9) | **139 (28,4)** |
| Recruiting | **30 (63,8)** | **9 (12,2)** | 0 (0) | 0 (0) | 0 (0) | **39 (28,1)** |
| Not recruiting | **15 (31,9)** | **12 (16,2)** | 0 (0) | 0 (0) | 1 (16,7) | **28 (20,1)** |
| Completed | 2 (4,3) | 53 (71,6) | 6 (100) | 2 (33,3) | 5 (83,3) | 68 (48,9) |
| Not stated | 0 (0) | 0 (0) | 0 (0) | 4 (66,7) | 0 (0) | **4 (2,9)** |
| Linked to record | 0 (0) | 20 (27,0) | 0 (0) | 2 (33,3) | 4 (66,7) | 26 (18,7) |
| Not Linked to record | **47 (100)** | 52 (70,3) | **6 (100)** | 3 (50,0) | **2 (33,3)** | **110 (79,1)** |
| Submitted to record | 0 (0) | 2 (2,7) | 0 (0) | 1 (16,7) | 0 (0) | 3 (2,2) |

ChiCTR, Chinese Clinical Trials Registry; IRCT, Iranian Registry for Clinical Trials; EU-CTR, European Clinical Trials Registry. All numbers are total counts. Percentages of the row "Total active interventional trials" refer to the total number of active trials (n = 490). Percentages in the section "Published trials per recruitment status" refer to the respective value of the row "Total" of the same section. All other percentages refer to the respective value of the row "Total active interventional trials". Data was extracted from the original dataset downloaded from the ICTRP and from the original registration record of each trial.

[1]Other registers include: JPRN, Japanese Primary Registry Network (n = 5); NTR, Netherlands Trial Registry (n = 2); ISRCTN, International Standard Registered Clinical Trial Number registry (n = 3); DRKS, German Clinical Trials Registry (n = 1); ReBEC, Brazilian Registry of Clinical Trials (n = 1); ANZCTR, Australian and New Zealand Clinical Trials Registry (n = 1); CTRI, Clinical Trials Registry India (n = 1).

*Trial records registered with the EU-CTR did not state the date of their last update, therefore no calculation was possible.

First, we checked all included trials for prospective or retrospective registration and found out that more than a third were registered retrospectively (37,2%). The highest rate of retrospectively registered trials had the IRCT with almost two thirds (64,9%), followed by the ChiCTR with almost half their trials being registered retrospectively (46,9%).

Then we checked each trial for their original recruitment status to compare it with the findings of our follow-up search after two years. Originally, more than half of the trials were registered as "recruiting" (55,1%) while the rest were registered as "not recruiting". As of the 11th of August 2022, the date of our follow-up search more than a third were still labelled as "recruiting" (39,2%), while slightly less than a third were either labelled as "not recruiting" (30,6%) or

were already labelled as "completed" (28,8%). The rest had not stated their recruitment status (1,4%). All 490 trials exceeded their date of first enrolment, which means all trials currently labelled as "not recruiting" had not been updated accordingly. 91,4% of the trials also exceeded their estimated or actual completion date of which more than a third (36,1%) were still labelled as "recruiting". Those trials therefore had not been updated accordingly as well. This adds up to a total of 66,7% trials which had not been updated accordingly.

Furthermore, we found out that nearly a quarter had not been updated once since their date of registration (24,3%). Out of those trials, 12,6% were labelled as "completed" in their latest version of their registration record, which implies that they were already registered as "completed" even though the downloaded database from the WHO stated otherwise.

Additionally, we found out that slightly less than half the trials which already published results were labelled as "completed" (48,9%) with the rest either been labelled as "recruiting", "not recruiting" or not having stated any status in decreasing order.

**Assessment of quality.** To assess the quality of the registration records and the registries itself we checked each record for completeness of the Trial Registration Data Set (TRDS) (S2 File, Table 6). Across all registers item 13 (interventions) was most frequent with 43,9% of all trials displaying insufficient information. Most of those trials have not stated any or insufficient information about the dosage, frequency of application or duration of the treatment. The EU-CTR had the highest rate of missing information on this item with 69,2% of their trials, followed by the ChiCTR, the IRCT and ClinicalTrials.gov. Item 21 (ethics review) was missing second most often with 40,2% of all trials not displaying any information on their ethics review. ClinicalTrials.gov had the highest rate of missing information on this item with 97,1% of their records. By not stating the allocation method, masking or the type of assignment, item 15 (study type) was missing third most often in 22,4% of all trial records. On this item the ChiCTR had the highest rate of missing information with 40,9% of their records. The summarization of results (item 23) was missing for 79,1% of the 139 trials which already published results by not providing a summary of the results or a link to the published results in their original record. ChiCTR and ICRT both had the highest rate of issues on this item with 100% of their published trials not providing a summary or a link to the publication on their record. This was followed by ClinicalTrials.gov with 70,3% and the EU-CTR with 50,0% of their trials. The contact information for public and scientific queries (items 7 and 8) were completely missing in 3,9% across all registers, however for another 15,9% of all trials, the information was only available in the WHO dataset and not in the original record from the source register. Seventy-seven of these trials were registered with ClinicalTrials.gov with a rate of 44,7% of their total 172 registered trials.

## Third objective: Assessment of research waste and redundancies

**Assessment of research waste.** For the assessment of research waste, we first investigated the discontinued trials for their reason of discontinuation. During the two-year follow-up process we identified that 17,4% of the interventional trials were discontinued due to several reasons. We sorted all stated reasons for discontinuation into 8 different categories. We also divided those trials by their source register and calculated the discontinued trials per total trials per register by dividing the total number of discontinued trials by the total number of trials from one register. 39,8% of the discontinued trials unfortunately did not state any reason. After that, the most frequent reason was "difficulties with enrolment" with 29,1% trials followed by "new evidence or new treatment guidelines" from other trials and/or publications with 12,6% trials. Divided into registers, ClinicalTrials.gov had the most discontinued trials with 56,3% of all discontinued trials followed by ChiCTR with 37,9% trials and EU-CTR with

**Table 6. Assessment of quality–Completeness of the trial registration data set.**

| Missing TRDS Items | Number of trials with missing or insufficient information sorted by registries (%) | | | | | |
| --- | --- | --- | --- | --- | --- | --- |
| | ChiCTR | ClinicalTrials.gov | IRCT | EU-CTR | Other[1] | Total |
| Total trials | 254 | 172 | 37 | 13 | 14 | 490 |
| | | | | | | |
| 4. Source(s) of monetary/material support | 2 (0,8) | 0 (0) | 0 (0) | 1 (7,7) | 2 (14,3) | 5 (1,0) |
| 5. Primary Sponsor | 0 (0) | 0 (0) | 0 (0) | 0 (0) | 4 (28,6) | 4 (0,8) |
| 7. Contact for public queries | | | | | | |
| Completely missing | 0 (0) | 19 (11,0) | 0 (0) | 0 (0) | 0 (0) | 19 (3,9) |
| Not shown on registry website | 0 (0) | 77 (44,8) | 0 (0) | 1 (7,7) | 0 (0) | 78 (15,9) |
| 8. Contact for scientific queries | | | | | | |
| Missing | 0 (0) | 19 (11,0) | 0 (0) | 0 (0) | 0 (0) | 19 (3,9) |
| Not shown on registry website | 0 (0) | 77 (44,8) | 0 (0) | 1 (7,7) | 0 (0) | 78 (15,9) |
| 11. Countries of recruitment | 0 (0) | 13 (7,6) | 0 (0) | 1 (7,7) | 0 (0) | 14 (2,9) |
| 13. Interventions | 155 (61,0) | 34 (19,8) | 15 (40,5) | 9 (69,2) | 2 (14,3) | 215 (43,9) |
| Intervention name described insufficiently | 42 (16,5) | 0 (0) | 0 (0) | 1 (7,7) | 0 (0) | 43 (8,8) |
| Dosage, frequency, or duration missing | 76 (29,9) | 4 (2,3) | 1 (2,7) | 8 (61,5) | 1 (7,1) | 90 (18,4) |
| Standard of care as comparator | 37 (14,6) | 30 (17,4) | 14 (37,8) | 0 (0) | 1 (7,1) | 82 (16,7) |
| 15. Study type | 104 (40,9) | 2 (1,2) | 0 (0) | 1 (7,7) | 3 (21,4) | 110 (22,4) |
| 16. Date of first enrolment | 0 (0) | 0 (0) | 0 (0) | 13 (100) | 0 (0) | 13 (2,7) |
| 18. Recruitment status | 0 (0) | 0 (0) | 0 (0) | 12 (92,3) | 0 (0) | 12 (2,4) |
| 21. Ethics review | 27 (10,6) | 167 (97,1) | 0 (0) | 0 (0) | 3 (21,4) | 197 (40,2) |
| 22. Completion date | 1 (0,4) | 0 (0) | 0 (0) | 0 (0) | 2 (14,3) | 3 (0,6) |
| 23. Summary results | * | * | * | * | * | * |
| 24. IPD sharing statement | 8 (3,1) | 37 (21,5) | 0 (0) | 12 (92,3) | 4 (28,6) | 61 (12,4) |

ChiCTR, Chinese Clinical Trials Registry; IRCT, Iranian Registry for Clinical Trials; EU-CTR, European Clinical Trials Registry. Only items which are missing information are displayed, all items not mentioned have not been missing information. All numbers are total counts. All percentages refer to the respective value of the row "Total trials". Data was extracted from the original registration records of each trial and analyzed according to S2 File.
[1]Other registers include: JPRN, Japanese Primary Registry Network (n = 5); NTR, Netherlands Trial Registry (n = 2); ISRCTN, International Standard Registered Clinical Trial Number registry (n = 3); DRKS, German Clinical Trials Registry (n = 1); ReBEC, Brazilian Registry of Clinical Trials (n = 1); ANZCTR, Australian and New Zealand Clinical Trials Registry (n = 1); CTRI, Clinical Trials Registry India (n = 1).
*Item 23 is represented in Table 5 in the section "Published trials per recruitment status".

12,6%. If we look at the rate of discontinued trials per total trials per register however, we see that out of all registers with more than 10 registered trials, the EU-CTR has the highest rate of discontinued trials with 50,0% followed by ClinicalTrials.gov with 21,8% and ChiCTR with 13,3% (Table 3).

We further evaluated potential research waste by analyzing the study design across the remaining 490 active interventional trials with extensive results shown in Table 7. Overall,

**Table 7. Assessment of research waste–Analysis of study design of active interventional trials.**

| | Study design (%) | | | | | | | |
|---|---|---|---|---|---|---|---|---|
| | **RCT** | **NRCT** | **Single Arm** | **RP** | **Sequential** | **NRP** | **Factorial** | **Total** |
| Total | 335 (68,4) | 48 (9,8) | 61 (12,4) | 20 (4,1) | 14 (2,9) | 8 (1,6) | 4 (0,8) | 490 (100) |
| **Masking** | | | | | | | | |
| None | 149 (44,5) | 17 (35,4) | 47 (77,0) | 11 (55,0) | 2 (14,3) | 5 (62,5) | 0 (0) | 231 (47,1) |
| Single | 26 (7,8) | 0 (0) | 0 (0) | 5 (25,0) | 0 (0) | 0 (0) | 0 (0) | 31 (6,3) |
| Double | 36 (10,7) | 2 (4,2) | 0 (0) | 0 (0) | 0 (0) | 0 (0) | 0 (0) | 38 (7,8) |
| Triple | 7 (2,1) | 0 (0) | 0 (0) | 1 (5,0) | 0 (0) | 0 (0) | 0 (0) | 8 (1,6) |
| Quadruple | 25 (7,5) | 0 (0) | 0 (0) | 1 (5,0) | 0 (0) | 0 (0) | 0 (0) | 26 (5,3) |
| N/A | 6 (1,8) | 24 (50,0) | 13 (21,3) | 0 (0) | 14 (100) | 3 (37,5) | 4 (100) | 64 (13,1 |
| Not Stated | 86 (25,7) | 5 (10,4) | 1 (1,6) | 1 (5,0) | 0 (0) | 0 (0) | 0 (0) | 93 (19,0) |
| | | | | | | | | |
| **Control** | | | | | | | | |
| None | 0 (0) | 0 (0) | 61 (100) | 20 (100) | 13 (92,9) | 7 (87,5) | 3 (75,0) | 104 (21,2) |
| Standard of Care | 263 (78,5) | 46 (95,8) | 0 (0) | 0 (0) | 1 (7,1) | 1 (12,5) | 1 (25,0) | 312 (63,7) |
| Placebo | 72 (21,5) | 2 (4,2) | 0 (0) | 0 (0) | 0 (0) | 0 (0) | 0 (0) | 74 (15,1) |
| **Study Phase** | | | | | | | | |
| 0 | 77 (23,0) | 14 (29,2) | 12 (19,7) | 1 (5,0) | 6 (42,9) | 1 (12,5) | 3 (75,0) | 114 (23,3) |
| 1 | 13 (3,9) | 3 (6,3) | 12 (19,7) | 2 (10,0) | 2 (14,3) | 1 (12,5) | 0 (0) | 33 (6,7) |
| 2 | 59 (17,6) | 9 (18,8) | 12 (19,7) | 3 (15,0) | 0 (0) | 2 (25,0) | 0 (0) | 85 (17,3) |
| 3 | 52 (15,5) | 2 (4,2) | 4 (6,6) | 7 (35,0) | 0 (0) | 1 (12,5) | 0 (0) | 66 (13,5) |
| 4 | 55 (16,4) | 7 (14,6) | 2 (3,3) | 4 (20,0) | 1 (7,1) | 1 (12,5) | 0 (0) | 70 (14,3) |
| N/A | 79 (23,6) | 13 (27,1) | 19 (31,1) | 3 (15,0) | 5 (35,7) | 2 (25,0) | 1 (25,0) | 122 (24,9) |

RCT, Randomized Controlled Trial; NRCT, Non-Randomized Controlled Trial; RP, Randomized Parallel Trial; NRP, Non-Randomized Parallel Trial. All numbers are total counts. Percentages of the row "Total" refer to the total number of active interventional trials (n = 490). All other percentages refer to the respective value of the row "Total". Data was extracted from the original dataset downloaded from the ICTRP and from the original registration record of each trial.

randomized-controlled trials (RCT) were most common with more than two thirds of all 490 interventional trials (68,3%), followed by single arm trials and non-randomized-controlled trials (NRCT). In the group of RCTs, 44,4% trials were not masked while 27,5% had not given any information about their masking. Seventy-eight-point-five percent of the RCTs used "standard of care" as the control while the rest used a placebo. In the group of NRCTs 35,4% trials were not masked and 10,4% did not state any information on their masking. Ninety-five-point-eight percent used "standard of care" and 4,2% trials a placebo as their control group. The most frequent study phase over all interventional trials was phase 0 with 23,3% followed by phase 2 trials with 17,3%. The study phase of 24,9% trials was either not applicable or not stated. The mean target size over all 490 trials was 591,56 subjects (SD 3445,83), the median 100 subjects, the 25 percentile 60 subjects and the 75 percentile 255,75 subjects. The target sizes according to each study phase are shown in Fig 5.

**Assessment of redundancies.** To identify possible redundancies among all interventional trials we analyzed the different therapeutic agents being investigated (Table 8). We identified a total of 261 trials which used a therapeutic agent other than complementary or alternative medicine and advanced medicinal products. Of those 261 trials 200 were still active and 61 discontinued. Over all 261 trials we identified 94 different drugs. (Hydroxy)-chloroquine was by far the drug tested most often with 31,8% trials. After that ritonavir, lopinavir and interferons

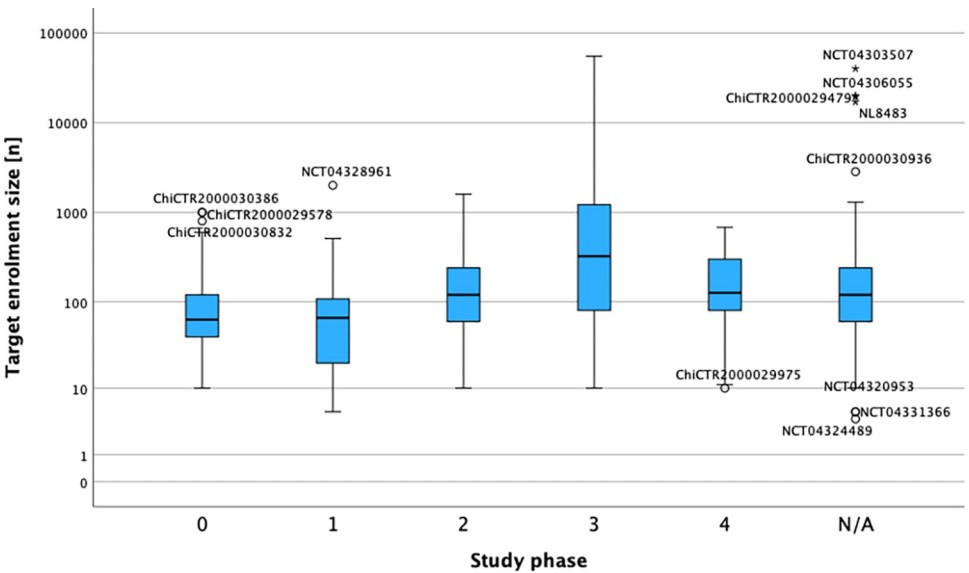

**Fig 5. Box-whiskers-plot of the target sizes per study phase of all active interventional trials.** X-axis: Active interventional trials separated into the different study phases; N/A, not applicable or not stated. Y-axis: Logarithmic representation of the target sizes. Outliers are titled with their individual trial-ID. Data was extracted from the original dataset downloaded from the ICTRP and from the original registration record of each trial.

were tested most often in decreasing order. (Hydroxy)-chloroquine also had the highest rate on discontinued trials with 38,6% out of 83 total trials. This was followed by remdesivir with 36,4% and azithromycin with 33,3%.

**Table 8. Assessment of redundancies–Analysis of investigated therapeutic agents across all interventional trials.**

|  | Active (%) | Discontinued (%) | Total (%) | Discontinued/Total in % |
|---|---|---|---|---|
| **Interventional trials** | 490 | 103 | 593 | 17,4 |
| **Therapeutic agents** |  |  |  |  |
| (Hydroxy)-Chloroquine | 51 (10,4) | 32 (31,1) | 83 (14,0) | 38,6 |
| Ritonavir | 35 (7,1) | 5 (4,9) | 40 (6,7) | 12,5 |
| Lopinavir | 30 (6,1) | 5 (4,9) | 35 (5,9) | 14,3 |
| Interferone | 24 (4,9) | 1 (1,0) | 25 (4,2) | 4,0 |
| Tocilizumab | 11 (2,2) | 4 (3,9) | 15 (2,5) | 26,7 |
| Favipiravir | 12 (2,4) | 0 (0) | 12 (2,0) | 0 |
| Azithromycin | 8 (1,6) | 4 (3,9) | 12 (2,0) | 33,3 |
| Remdesivir | 7 (1,4) | 4 (3,9) | 11 (1,9) | 36,4 |
| Corticosteroids | 7 (1,4) | 3 (2,9) | 10 (1,7) | 30,0 |
| Arbidol | 6 (1,2) | 2 (1,9) | 8 (1,3) | 25,0 |
| Vitamin C | 2 (0,4) | 3 (2,9) | 5 (0,8) | 60,0 |
| Oseltamivir | 5 (1,0) | 0 (0) | 5 (0,8) | 0 |
| Losartan | 4 (0,8) | 1 (1,0) | 5 (0,8) | 20,0 |
| ACEs/ARBs | 2 (0,4) | 2 (1,9) | 4 (0,7) | 50,0 |
| Ribavirin | 4 (0,8) | 0 (0) | 4 (0,7) | 0 |

All numbers are total counts. Percentages refer to the respective value of the row "Interventional trials". Data was extracted from the original database downloaded from the ICTRP and the original registration record of each trial.

## Discussion

### Key results and interpretation

**First objective: Assessment of spatial and temporal evolution of clinical research of COVID-19 across the globe.** Our key findings for this objective were: 1) The overall research activity was much higher than in any pandemic, epidemic or health crisis before. 2) China was the country with the most registered trials in total and per confirmed cases, however in terms of trials per population size, the Netherlands and France were leading. Looking at the evolution of the numbers from each country though, it appears that the number of trials per cases and population will align over time. 3) Our results suggest that declarations and warnings from institutions like the WHO did not have that much of an impact on the emergence of clinical research. Rather, clinical research might had been triggered by panic due to rising disease incidence.

1) Our initial sampling goal of 1000 registered trials was reached after only approximately 3 months while the current total number of registered trials (accessed 29[th] of June 2023) in the ICTRP related to COVID-19 was 15834. For other pandemics and epidemics, the total numbers of registered trials were significantly lower, e.g., H1N1 (n = 509), Ebola (n = 140), SARS (n = 4053), MERS (n = 21) and even HIV/Aids (n = 11764). To put this into perspective, H1N1, Ebola and MERS combined did not have as many registered trials as we found for COVID-19 in the first 3 months [14].

2) The temporal and spatial development of clinical research in the COVID-19 pandemic in general followed the spread of the disease, which was expected. China was the most dominant country with more than two thirds of all trials originating from there and leading on the number of trials per 100 cases with 0,806 vs. Japan being second with 0,512 trials per 100 cases. Considering trials per population though, China was not as dominant as the previous numbers might imply with 0,465 trials per 1 million inhabitants vs. Netherlands with 0,8 and France with 0,682. Another trial which examined the characteristics of published scientific articles in the first 3 months of the COVID-19 outbreak came to similar results with China having published more than half of the articles with 50,5% and 0,422 articles per 1 million inhabitants [15]. However, in terms of articles per 100 disease cases, our study shows different results. In our study China registered third most trials per cases while China was in the bottom end of articles per cases in the opposed study. The opposed study took the number of confirmed cases from the 2[nd] of March 2020, before the COVID-19 situation was declared as a pandemic, while our designated date was set one month later, on the 1[st] of April 2020, and after the declaration, which could explain the differences. Thus, in our study, the number of confirmed cases per country was significantly higher, especially in countries other than China due to the vast spread of the disease in March 2020, and thereby the differences in terms of trials per cases are smaller. Our results suggest that the numbers of trials per cases will continue to align over time, and thus China will only be standing out in the first few months of the COVID-19 outbreak.

3) Nevertheless, the fact that only 4,9% of all trials before the pandemic declaration by the WHO originated from outside of China was still surprising, since the outbreak was declared "a public health emergency of international concern" one month prior to being officially declared a pandemic. One would have expected a more robust response at this time [1]. The graphical development of the cumulative cases and trials per date of each region could be an explanation for this (Fig 3). We see that in all countries except China the number of cases began to increase dramatically only a few days before the declaration, which was then followed by a rapid growth of registered trials. In China on the other hand, the number of cases stagnated around the same time which was then followed by a flattening curve of registered trials. All in all, this

raises the question of whether official declarations by the WHO or other institutions are influencing the start of research or whether panic due to the individual case numbers in each country are the main driver for research. Our findings, however, imply the latter.

**Second objective: Assessment of transparency and quality–trial registration.** Our key findings for this objective were: 1) There is a lack of transparency in trial registration expressed by retrospective registration and registration records not kept up to date with conflicting and invalid information being displayed. 2) The overall quality of trial registration seems to have improved since the introduction of the Trial Registration Data Set, yet there is still crucial information missing.

Trial registration is a crucial instrument which aims to ensure transparency, quality and validity of research and therefore prevent publications bias, reproducibility bias, and selective reporting bias [8, 16]. Transparency of trial registration was defined as maintaining the trial registration records and updating them in a timely manner, to ensure that every information extracted from these records can be considered as valid and up to date. The quality of trial registration was assessed by screening each registration record for prospective registration and for completeness of the trial registration data set.

1) Prospective registration is an important tool to prevent selective reporting and publication bias. By making registration records a public matter before recruiting the first patient, it is very unlikely to alter the data to fit the results since every change to the protocol or record is publicly accessible [8, 16, 17]. This topic is also stated in the declaration of Helsinki with the words "Every research study involving human subjects must be registered in a publicly accessible database before recruitment of the first subject" [18]. The ICMJE therefore requires prospective registration from every clinical trial initiated after July 1, 2005 in order to be considered for publication in any of their member journals [19]. Though, several studies concluded that retrospective registration and thus the possibility of publication and selective reporting bias is still present [20, 21]. One study, which analyzed all trials registered in the ANZCTR from 2006 to 2015, came to the result that the number of prospectively registered trials increased from 42% in 2006 to 63% in 2012 and plateaued from thereon [22]. Our study came to similar results with one third of all trials being registered retrospectively. Particularly problematic is the fact that we identified 15 trials which were registered retrospectively after the recruitment was already completed. Although other studies suggest that the number of trials which are being registered retrospectively appear to decrease, there is still a lot of work to do. If retrospective registration exists, selective reporting and publication bias will be a reoccurring problem. Registries and especially journals are in need follow the ICMJE rules more strictly by no longer accepting and publishing studies that were registered retrospectively to overcome this issue.

Keeping trial records up to date is another very important aspect of trial registration. To enforce this, all primary and partner registries in the WHO Registry Network signed a form which endeavors to keep registered information up-to-date, some registries even implemented reminder systems to facilitate this by notifying the authors at least every 6 months if there was no update [23]. To our knowledge this is the first study which conducted a follow-up to track the progression of trial registration records over time to evaluate the accuracy and validity of the information being displayed. We discovered that two thirds of the trials haven't been updated accordingly and a quarter of all trials have never been updated after two years since their registration. Particularly unfortunate is the fact that more than half of all trials which already published results were not even labelled as completed. One of the goals of trial registration is that the public and scientific community can consider every information and data displayed as accurate and valid. If this is not ensured for arguably one of the most important and

fundamental information in trial registration, i.e., the recruitment status, the whole concept of trial registration is to be questioned.

2) The Trial Registration Data Set (TRDS), which was designed by the WHO and is required by the ICMJE, represents another tool to ensure transparency, quality and validity in clinical research and trial registration by providing a minimum set of items each trial record has to display in order to be considered adequate and eligible for publication [24]. To ensure this and to further conquer publication and reproducibility bias, it is crucial that the information of several key items (i.e. interventions, contact information, study type) is informative and complete. Though, our results showed that there were several items with a high rate of missing or insufficient information, e.g., Item 13 (interventions) with 43,9%, Item 21 (ethics review) with 40,2% and Item 15 (study type) with 22,4%. Also, only 26/139 (18,7%) trials displayed Item 13 (summary results) sufficiently and either provided a link or a summarization for already published results. We also detected differences between the registries on which items are missing most often. The ChiCTR for example displayed insufficient information for item 13 (interventions) in 61% and for item 15 (study type) in 40,9% of all their records, while ClinicalTrials.gov was missing Item 21 (ethics review) in 97,1% of their registration records and item 7 and 8 (contact for public and scientific queries) in 44,8%. One study, which took a random sample of 439 records registered between 2008 and 2009 from the ICTRP and assessed those in terms of their contact, intervention and primary outcome information, found that 32,3% of the trials have not provided sufficient contact information and only 44,2% provided sufficient information on interventions [25]. Another study conducted by the same authors 4 years later with the same design found that 15,1% of the trials have not provided sufficient contact information and only 51,9% provided sufficient information on interventions [26]. In our study only 3,9% trials were completely missing the contact information of which all were registered on ClinicalTrials.gov. The differences regarding the contact information can be explained with the fact that in our study ClinicalTrials.gov is underrepresented with 35,1% of all trials being registered on this register compared to 85,9% respectively 57,5% in their studies. Results from another study in 2009 which assessed the whole data set (at that time only 20 items) showed a much higher rate of missing information for nearly every item [27]. Since these studies depict rather the early years after trial registration was implemented, our findings also suggest that there was an overall improvement in registration quality over time, yet some key items like interventions have been and still are missing critical information. These suggestions meet the results from two recent studies which assessed the quality of trial records investigating Traditional Chinese Medicine. The authors concluded that especially the complex items like interventions, study design and primary outcomes were missing crucial information [28, 29]. Although, the implementation of trial registration and therefore compliance with the TRDS was advocated over 17 years ago, there is evidently still a huge lack of quality and transparency. To conquer publication bias, selective reporting bias and reproducibility bias, institutions like the WHO and the ICMJE, must work more closely with the registers, journals, and researchers to overcome these issues. As a possible solution for the differences between registers, lack of quality in trial registration and to ensure that the information is displayed equally, we suggest a unified item mask which each register must implement to be considered as a partner for the WHO Registry Network and be eligible for publication. This, however, must be enforced by the WHO and the ICMJE, as there is no need for registers to change if their trials are still being published.

**Third objective: Assessment of research waste and redundancies.** Our key findings for this objective are: 1) The early stages of clinical research about COVID-19 showed clear signs of research waste. Furthermore, research waste appears to be a reoccurring problem in the scientific world. 2) "Hype train phenomenon" in early stages of research assumingly due to a lack

of dedicated supervising institutions and coordination which leads to unnecessary redundancies and ultimately to more research waste.

Research waste and redundancies in clinical research are significant problems that can have serious implications for the scientific community, as well as for the patients who participate in clinical trials. Research waste refers to the unnecessary use of resources including time, money, and other valuable resources. Research waste will therefore be measured by the rate of discontinuation and the quality of study design. Redundancies, on the other hand, occur when multiple studies are conducted on the same or similar topics, resulting in duplication of efforts and resources. To assess this issue, we screened all drug trials for their therapeutic agents to depict potential redundancies which would also lead to more research waste.

1) Discontinuation displays one source for wasting money, time, and other valuable resources [30–32]. Nearly 2 years after the last trial of our study was registered, we identified 17,3% discontinued interventional trials. The most common reason for discontinuation was difficulties with enrolment of patients with 29,1% trials. Discontinuation rates ranging from as low as 11% and up to 43% had been displayed in several other studies. Similar to our findings, the most common reason for discontinuation in all of those trials was difficulties with enrolment [30, 31, 33–37]. However, a systematic review from 2016 suggested that most reasons for recruitment failure could be anticipated in the planning phase of a trial and thus are preventable. The authors also provided a checklist to ensure that weak spots in study designs are being identified [38]. Since transparency is important to detect errors in the planning of studies and to prevent them in the future, it is even more problematic if no reason for discontinuation is given. Unfortunately, 39,8% had not stated any reason for discontinuation. Even in discontinued trials, transparency is important to detect errors in the planning of trials and prevent those mistakes in the future by learning from one another.

Another important aspect to avoid research waste is an appropriate and robust study design. Although, more than two thirds of all interventional trials are designed as RCTs, which are mostly ranked on top of the evidence levels and hailed as the "gold-standard" of evidence-based-medicine, almost one third of trials used other designs, often without any control, which is ranked on the bottom of the evidence pyramid [39]. Further, even the RCTs have flaws in their design as nearly half of them were not masked which can lead to several bias such as selection bias, which was supposed to be eliminated by the randomization itself [40]. We speculate that panic and the willingness to help may have driven these "short-cuts" in robustness.

2) To further avoid research waste, it is also important to create awareness for redundancies in clinical research. Trial registration was intended as a tool to diminish this by making planned and ongoing research available for the public and scientific world.

(Hydroxy)-chloroquine (HCQ/QC), as the most prominent example, gained huge popularity in the early months of COVID-19. However, the timeline of HQC/QC showed contradicting information about the pros and cons of its therapeutic use against COVID-19 which was also influenced by the media and even leading politicians in effected countries [41]. The therapeutic use was halted when the US-FDA revoked its emergency authorization on June 15, 2020 [42]. Early reviews from 2020 supported this by indicating weak and conflicting evidence about the efficacy and safety of HQC/CQ [43–46]. Throughout this evolution, there was always an urgent call for better designed and properly powered trials to gain unequivocal data safety and efficacy for HCQ/CQ usage in COVID-19 [44, 47]. Though, our results show that 14% of all interventional trials investigated the usage HQC/CQ with more than a third (38,6%) of those trials being discontinued before completion. This is not only an example for research waste and redundancies but also raises the question on whether numerous trials assessing the same topic operationally fragmented in isolated silo-structures fulfils the purpose of gaining

more evidence. The given tools (e.g., trial registration and the registers itself) must be used more thoroughly to detect similar trials and encourage collaboration instead of conducting own trials one after the other without any coordination. This could not only help with the power of trials but could also lower the discontinuation rates by preventing recruitment failure through multicentric designs and thus also prevents unnecessary exposure to clinical trials for patients.

## Limitations

The limitations of this study are the following:

1. Sine the registered trials are updated on a regular weekly basis the data at the point of download from the WHO ICTRP search portal can be considered as up to date as of the 22$^{nd}$ of April, meaning all trials registered to this date were considered for inclusion in this study. Nevertheless, we cannot rule out with complete certainty that no additional trials were added afterwards which could lead to a different number of included trials if the dataset was downloaded on another date. Also, we expect the data set to be different if downloaded now, since the trial registration protocols can be updated and changed and therefore the information displayed in the data set by the WHO will also be changed.

2. We only used the keyword "COVID-19" as a search criterion which could have biased our search outcome.

3. Since our objective was to observe and analyze the first surge of clinical research and our goal of 1000 registered trials was already exceeded on the 1$^{st}$ of April 2020, we only cover a small period. Our findings therefore only describe the early stages of this pandemic, i.e., the initial "chaos phase" and are definitively not generalizable to the entire pandemic.

4. As this study only evaluates the quality and validity of the registration records, we do not presume to make any statement about the ultimate quality of the trials itself. Further studies must assess whether the quality deficiencies we have identified in the records are carried through to publication.

5. Including a larger number of clinical trials would have allowed a better inside but we deliberately chose to only assess the early stages of such a situation. It would be interesting to investigate if the results of this analysis changed along the pandemic.

## Conclusion

To our knowledge, this is the first study which examined the quality of trial registration in the early "chaos phase" of the COVID-19 pandemic and the first study which conducted a follow-up to assess the development of the recruitment status of registration records. The temporal and spatial evolution of clinical research turned out as expected. China, being the country the virus originated from, was the first country to start clinical research and was overall the country with the most registered trials in the first months of the pandemic. The spread of the disease was also accompanied by emerging trials from other countries, and it appears that official pandemic declarations from institutions like the WHO have less impact on the uprising of research than the number of cases in each country. While the information of trial records is easily available and accessible for everybody, we detected a huge lack of quality and validity, which is not due to the pandemic situation but rather a general problem. With the data we found and the experience we gathered while examining the trial records, we highly doubt the meaningfulness and benefit of trial registration in its current form, especially for rapidly

evolving situations across global geographies and language zones. If the information is incomplete, not up-to-date and registered retrospectively, the usefulness is not given. Registries will therefore only be used for mandatory registration and thus are deprived from other important functions: 1) facilitating recruitment by creating awareness for researchers and possible participants about recruiting trials 2) enabling and improving collaboration between researchers 3) identifying gaps in clinical research 4) avoiding redundancies by identifying similar trials 5) reducing research waste by identifying potential problems in study design. Especially in the "scientific chaos phase" of an emerging pandemic like COVID-19, these functions can be of tremendous help by being an organizer of chaos and therefore reducing information overload and research waste. Advocating and emphasizing these functions can furthermore help in the early stages of future pandemics and epidemics by preventing chaos and panic before it even emerges. An important mission of the WHO is to break the "panic-then-forget" cycle in future health crisis [48]. The scientific response proved to be intact. Many colleagues all over world fought the pandemic with dedication, bravery, and sacrifices, but as Dr. Mike Ryan from the WHO stated: "Preparation is the most active act in public health, not response. Preparation is response, and we make a false separation between these concepts" [48, 49]. Scientific communication and coordination tools such as clinical trials registry tools—if kept current and accurate—are valuable instruments to strengthen societal resiliency in global, medical disaster situations. Appropriate disaster preparation implies that these systems be optimized—taking our findings and suggestions for improvement into account—before the next disaster strikes [50, 51].

We believe that the WHO in cooperation with the ICMJE can have the most impact on improving quality and usefulness of trial registration as they define the requirements for registration and publication.

## Supporting information

**S1 File. STROBE-statement checklist.**
(PDF)

**S2 File. 22 items from the original data set.**
(PDF)

**S3 File. Trial registration data set items.**
(PDF)

**S1 Dataset.**
(SAV)

**S2 Dataset.**
(XLSX)

**S3 Dataset.**
(SAV)

**S4 Dataset.**
(SAV)

## Acknowledgments

This research article is part of Till Adamis doctoral thesis.

## Author Contributions

**Conceptualization:** Till Adami, Markus Ries.

**Data curation:** Till Adami.

**Formal analysis:** Till Adami.

**Investigation:** Till Adami, Markus Ries.

**Methodology:** Till Adami, Markus Ries.

**Project administration:** Till Adami, Markus Ries.

**Resources:** Till Adami, Markus Ries.

**Software:** Till Adami.

**Supervision:** Markus Ries.

**Validation:** Till Adami.

**Visualization:** Till Adami, Markus Ries.

**Writing – original draft:** Till Adami.

**Writing – review & editing:** Till Adami, Markus Ries.

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
