## [Decision Letter · Decision Letter 0]

18 Sep 2023

PONE-D-23-21783The scientific chaos phase of the Great Pandemic: A longitudinal analysis and systematic review of the first surge of clinical research concerning COVID-19PLOS ONE

Dear Dr. Adami,

Thank you for submitting your manuscript to PLOS ONE. After careful consideration, we feel that it has merit but does not fully meet PLOS ONE’s publication criteria as it currently stands. Therefore, we invite you to submit a revised version of the manuscript that addresses the points raised during the review process.

We look forward to receiving your revised manuscript.

Kind regards,

Ashraful Hoque, MD

Academic Editor

PLOS ONE

Journal Requirements:

“Markus Ries is a guest editor of the call for papers “Prediction and Mitigation of Natural Hazards”.”

Reviewers' comments:

Reviewer's Responses to Questions

**Comments to the Author**

1. Is the manuscript technically sound, and do the data support the conclusions?

Reviewer #1: Yes

Reviewer #2: Partly

2. Has the statistical analysis been performed appropriately and rigorously? 

Reviewer #1: Yes

Reviewer #2: I Don't Know

3. Have the authors made all data underlying the findings in their manuscript fully available?

Reviewer #1: Yes

Reviewer #2: Yes

4. Is the manuscript presented in an intelligible fashion and written in standard English?

Reviewer #1: Yes

Reviewer #2: Yes

5. Review Comments to the Author

Reviewer #1: Dear Authors, this is an interesting manuscript, but the Tables 1-6 are cutoff and thus not accessible/readable in the PDF file that I have received for review. Please reformat these items for further review.

Reviewer #2: This manuscript has great potential to provide some insight into research patterns during the COVID-19 pandemic. The authors explained their methods in great detail, demonstrating the great care they took in identifying both intervention and observational studies.

The authors demonstrated some very important research trends in this study and I think that the comparison to research in other pandemics and high-risk infectious diseases (eg, HIV) was important. I think the authors provide some sound advice to researchers and their research suggeests that there has been a great deal of "spinning wheels" as well as some poor quality research churned out, so I like that this paper takes the perspective that we need some caution with COVID-19 research.

A few thoughts for improvement - one thing that was difficult for me was the description of the methods. The methods were very detailed, but a bit long. Are there ways to streamline this? Can the search terms be put into a table, perhaps? That would help the readability of the manuscript and make it easier to follow. I found myself re-reading the methods again and again to be sure I understood what was done. So a bit of streamline could be helpful here.

I had tremendous difficulty with the tables, and honestly, it was a barrier to my providing my review. It was so hard to provide feedback when I couldn't read the full tables. Details are here:

In the uploaded manuscript, table 1 was cut off after DEU column and I was not able to fully assess this table.

In the uploaded manuscript, table 2 was cut off after the "other" column, so I was not able to fully assess this table.

In the uploaded manuscript, table 3 as cutt off after the "other" column, so I was not fully able to assess this table.

In the uploaded manuscript, table 4 was cut off after the EU column, so I was not able to fully assess this table.

In the uploaded manuscript, table 5 was cut off after the EU-CTR column, so I was not fully able to assess this table.

In the uploaded manuscript, table 6 was cut off after the "sequential" column; the column after that starts with the letter N but I could not see any more of this table and therefore I was not able to fully assess it.

I think the issue is that once the Word (or whatever you used to write the manuscript) was converted to a PDF, the formatting of the tables was thrown off and everything was cut off. So that was very hard for me to give feedback on. I am uploading the PDF so you can see how they are cut off.

For the graphs by country, is there any way to have one giant graph of all of the countries together, with different colors representing them? There are just so many tables and a lot to look at. A unified table with the countries together could be instuctive and interesting for comparison.

I hope this is helpful and many apologies for my delayed review.

6. PLOS authors have the option to publish the peer review history of their article (what does this mean?). If published, this will include your full peer review and any attached files.

Reviewer #1: No

Reviewer #2: No

---

## [Author Response · Author response to Decision Letter 0]

27 Oct 2023

Journal requirements

Comment 1:

Thank you for stating the following in the Competing Interests section:

“Markus Ries is a guest editor of the call for papers “Prediction and Mitigation of Natural Hazards”.”

Answer:

We included the updated competing interest statement in the cover letter.

Reviewer #1

Comment 1:

Dear Authors, this is an interesting manuscript, but the Tables 1-6 are cutoff and thus not accessible/readable in the PDF file that I have received for review. Please reformat these items for further review.

Answer:

In our initial submission we strictly complied with the submission guidelines, which state that tables should not be resized and shaped to fit the margins of the manuscript page. Tables that run of those margins can be seen using the “Draft View” of Microsoft Word. Unfortunately, it seems the reviewers have received an automatically created PDF version instead of the Microsoft Word file of the manuscript. We have immediately contacted the PLOS ONE staff team directly, asking for a technical resolution, but unfortunately have not received any answer or solution for this issue so far. To quickly facilitate your important review, we have adjusted all tables to fit the margins of the manuscript page. However, the tables now look very squashed and not as we would like to see in a published version.

Reviewer #2

Comment 1:

This manuscript has great potential to provide some insight into research patterns during the COVID-19 pandemic. The authors explained their methods in great detail, demonstrating the great care they took in identifying both intervention and observational studies.

The authors demonstrated some very important research trends in this study and I think that the comparison to research in other pandemics and high-risk infectious diseases (eg, HIV) was important. I think the authors provide some sound advice to researchers and their research suggests that there has been a great deal of "spinning wheels" as well as some poor quality research churned out, so I like that this paper takes the perspective that we need some caution with COVID-19 research.

Answer:

We would like to express our gratitude for this comment and are extremely pleased for this encouraging statement. We could not agree more.

Comment 2:

A few thoughts for improvement - one thing that was difficult for me was the description of the methods. The methods were very detailed, but a bit long. Are there ways to streamline this? Can the search terms be put into a table, perhaps? That would help the readability of the manuscript and make it easier to follow. I found myself re-reading the methods again and again to be sure I understood what was done. So a bit of streamline could be helpful here.

Answer:

Thank you for this important comment. We totally agree with the reviewer and changed materials and methods to streamline this part. We changed the column of table 1 from “instruments” to “Data sources” and added a new column “variables”. All important information about which data was used on which objective can now be seen in one table. We also added an additional supporting file (S2) with the 22 items which were downloaded from the WHO in the original data set to prevent unnecessary listing of items. We feel that maintaining the reproducibility is very important, which is why we refrained from further streamlining of the methods.

Comment 3:

I had tremendous difficulty with the tables, and honestly, it was a barrier to my providing my review. It was so hard to provide feedback when I couldn't read the full tables. Details are here:

In the uploaded manuscript, table 1 was cut off after DEU column and I was not able to fully assess this table.

In the uploaded manuscript, table 2 was cut off after the "other" column, so I was not able to fully assess this table.

In the uploaded manuscript, table 3 as cutt off after the "other" column, so I was not fully able to assess this table.

In the uploaded manuscript, table 4 was cut off after the EU column, so I was not able to fully assess this table.

In the uploaded manuscript, table 5 was cut off after the EU-CTR column, so I was not fully able to assess this table.

In the uploaded manuscript, table 6 was cut off after the "sequential" column; the column after that starts with the letter N but I could not see any more of this table and therefore I was not able to fully assess it.

I think the issue is that once the Word (or whatever you used to write the manuscript) was converted to a PDF, the formatting of the tables was thrown off and everything was cut off. So that was very hard for me to give feedback on. I am uploading the PDF so you can see how they are cut off.

Answer: 

Please see our answer to Reviewer #1, comment 1. We altered the tables and they should now appear fully within the margins of the page. In addition, we are in contact with the PLOS ONE staff to identify the best possible technical solution for this technical conversion issue.

Comment 4:

For the graphs by country, is there any way to have one giant graph of all of the countries together, with different colors representing them? There are just so many tables and a lot to look at. A unified table with the countries together could be instructive and interesting for comparison.

Answer: 

Thank you again for this important and helpful critique. We agree with the reviewer and have created one giant graph. Giving each country a different colour would have resulted in a convoluting 20 different colours. As an alternative we decided to work with two colours instead, blue representing the cumulative cases of COVID-19 and orange representing the cumulative trials. As the main message of this graph is to highlight the differences in cases and trial evolution between China and the rest of the world, we highlighted the Chinese graphs and dashed all other graphs. We are convinced that the new graph allows a much better comparison than the previous graphs.

---

## [Editor Report · Decision Letter 1]

17 Nov 2023

The scientific chaos phase of the Great Pandemic: A longitudinal analysis and systematic review of the first surge of clinical research concerning COVID-19

PONE-D-23-21783R1

Dear Dr. Adami,

We’re pleased to inform you that your manuscript has been judged scientifically suitable for publication and will be formally accepted for publication once it meets all outstanding technical requirements.

Kind regards,

Ashraful Hoque, MD

PLOS ONE
---

## [Editor Report · Acceptance letter]

22 Nov 2023

PONE-D-23-21783R1 

The scientific chaos phase of the Great Pandemic: A longitudinal analysis and systematic review of the first surge of clinical research concerning COVID-19 

Dear Dr. Adami:

I'm pleased to inform you that your manuscript has been deemed suitable for publication in PLOS ONE. Congratulations! Your manuscript is now with our production department. 

Kind regards, 

on behalf of

Dr. Ashraful Hoque 

Academic Editor

PLOS ONE